# Luck of the draw: Role of chance in the assignment of medicare readmissions penalties

**Andrew D. Wilcock**[1]*, **Sushant Joshi**[2,3], **José Escarce**[4], **Peter J. Huckfeldt**[5], **Teryl Nuckols**[6], **Ioana Popescu**[4], **Neeraj Sood**[2,3]

**1** University of Vermont College of Medicine, Burlington, Vermont, United States of America, **2** Sol Price School of Public Policy, Department of Health Policy and Management, University of Southern California, Los Angeles, California, United States of America, **3** Schaeffer Center for Health Policy and Economics, University of Southern California, Los Angeles, California, United States of America, **4** Department of General Internal Medicine, University of California, Los Angeles, California, United States of America, **5** School of Public Health, Division of Health Policy and Management, University of Minnesota, Minneapolis, Minnesota, United States of America, **6** Division of General Internal Medicine, Department of Medicine, Cedars-Sinai Medical Center, Los Angeles, California, United States of America

* andrew.wilcock@med.uvm.edu

**Data Availability Statement:** The data underlying the results presented in the study are available from Centers for Medicare and Medicaid Services

## Abstract

Pay-for-performance programs are one strategy used by health plans to improve the efficiency and quality of care delivered to beneficiaries. Under such programs, providers are often compared against their peers in order to win bonuses or face penalties in payment. Yet luck has the potential to affect performance assessment through randomness in the sorting of patients among providers or through random events during the evaluation period. To investigate the impact luck can have on the assessment of performance, we investigated its role in assigning penalties under Medicare's Hospital Readmissions Reduction Policy (HRRP), a program that penalizes hospitals with excess readmissions. We performed simulations that estimated program hospitals' 2015 readmission penalties in 1,000 different hypothetical fiscal years. These hypothetical fiscal years were created by: (a) randomly varying which patients were admitted to each hospital and (b) randomly varying the readmission status of discharged patients. We found significant differences in penalty sizes and probability of penalty across hypothetical fiscal years, signifying the importance of luck in readmission performance under the HRRP. Nearly all of the impact from luck arose from events occurring after hospital discharge. Luck played a smaller role in determining penalties for hospitals with more beds, teaching hospitals, and safety-net hospitals.

## Introduction

Pay-for-performance programs are one strategy used by health plans to improve the efficiency and quality of care delivered to beneficiaries. Under such programs, providers are often compared against their peers in order to win bonuses or face penalties in payment. Yet luck has the potential to impact payers' ability to make accurate and fair assessments of performance.

under a Data Use Agreement. For access, please see ResDAC for details (https://resdac.org/).

**Funding:** Supported by grants from The Agency for Healthcare Research and Quality (R01 HS024284) and National Institute of Aging (R01AG046838) (all authors). The funders had no role in study design, data collection and analysis, decision to publish, or preparation of the manuscript.

**Competing interests:** The authors have declared that no competing interests exist.

Medicare, the national health insurance program in the United States for Americans aged 65 years and older, has developed and employed a suite of pay-for-performance initiatives since the passage of the Affordable Care Act in 2010.

Intended to improve the quality of care during and shortly after hospitalization, Medicare's Hospital Readmissions Reduction Program (HRRP) has penalized hospitals for having "excess" 30-day readmissions since fiscal year (FY) 2013 [1]. Excess readmissions exist when a hospital's risk-adjusted readmission rate during a three-year look-back period exceeds the rate that would be expected based on its patients' clinical characteristics, age, and sex. Despite the intent of the HRRP, readmissions are a function of not only quality of care but also of myriad factors that are omitted from HRRP risk-adjustment methods, outside of a hospital's control, and affected by luck—i.e., random variation [2, 3]. In other healthcare contexts, random variation in outcome measures has reduced precision, leading to poor reliability when classifying providers based on performance [4]. Little is known about the role of luck in determining whether a hospital will be penalized under the HRRP or in determining the size of a hospital's penalty.

Luck has the potential to affect hospital readmission rates primarily through two mechanisms. Leading up to the index admission, random events—like who gets sick in a hospital's catchment area or decisions by ambulance drivers—influence who is admitted to the hospital during any particular time interval. Risk adjustment does not fully account for unmeasured characteristics, such as income and frailty, among many others. Thus, randomness in who is admitted to the hospital may lead to some degree of randomness in readmission risks. During the 30 days after hospital discharge, randomness from events unrelated to the index hospitalization, such as from exacerbations of different clinical conditions, accidental injuries, and instability in essential social supports, also affect readmissions. The degree to which randomness affects readmission rates likely differs across hospitals based on attributes of the hospital or its patient base. For instance, larger hospitals may be less subject to randomness, as may hospitals whose patient base is more homogeneous.

We evaluated the role of luck in the assignment of HRRP penalties by answering three related questions. First, we quantified the overall extent to which random variation determines a hospital's penalty status and the size of any penalties in any given year. Our analysis accounted for variation due to which patients are admitted to a hospital and from events occurring after the index discharge. Second, we isolated the degree to which readmission rates among patients admitted to a specific hospital are influenced by random variation in events after the index discharge. Third, we examined the association between hospital characteristics such as size, ownership, teaching status, urban/rural location, region of the U.S., safety net status, and the proportion of a hospital's admissions comprised of Medicare beneficiaries (Medicare share) and the degree to which randomness influences HRRP penalties. For example, smaller hospitals or hospitals with lower Medicare share may have fewer patients eligible for the HRRP which would inherently be subject to greater variation.

## Methods

### Overview

To investigate the role of luck in the assignment of HRRP penalties, we performed simulations that estimated a hospitals' FY2015 readmission penalties in 1,000 different hypothetical fiscal years. These hypothetical fiscal years were created by: (a) randomly varying the patients admitted to a hospital and (b) randomly varying the readmission status of patients conditional on hospital discharge. We described the variation in penalties within a hospital across these 1,000 different hypothetical fiscal years. Greater variation in penalties within a hospital implies a

greater role of luck or chance in determining penalties, since penalties varied across hypothetical fiscal years due only to random variation in patients admitted and their readmission status. We also examine the discordance between FY2015 penalties and penalties in hypothetical fiscal years. Finally, we examine which hospital characteristics are associated with the role of luck in determining readmission penalties.

## Setting and study population

Study hospitals included 3,130 acute care hospitals that were subject to the HRRP in FY2015. The underlying study population included 4,159,463 discharges from adults age 65 or older who had an index discharge between July 2010 and June 2013 for one of five conditions targeted by the HRRP: acute myocardial infarction (AMI), heart failure (HF), pneumonia, chronic obstructive pulmonary disease (COPD), and elective total hip and knee arthroplasty (THA and TKA). To qualify as an index discharge, the beneficiary had to be discharged alive into the community (i.e., not transferred to a post-acute care facility) and had to be continuously enrolled in Medicare Parts A and B (i.e., coverage for inpatient and outpatient/professional services respectively) from 12 months prior to the end of their admission, and continuously enrolled in Part A through the post-discharge 30 day evaluation window for readmission [5, 6]. Patients that transferred between hospitals during their indexed admission are attributed to the final hospital that discharged the patient. Medicare used discharges from this three-year period to assign penalties in FY2015.

## Data sources

Information on hospitalizations and patient comorbidities came from the 2010–2013 Medicare Provider and Analysis Review (MEDPAR) files. We used information on patient demographics and Medicare enrollment information from the corresponding Master Beneficiary Summary files. We obtained hospital characteristics from the Medicare Impact files and the 2012 American Hospital Association Annual Survey. Hospital characteristics were missing for 3.8% of our study sample.

## Measures

The primary outcome was a dichotomous indicator (yes/no) for 30-day all-cause unplanned readmission, per program specification. Multiple unplanned readmissions during the 30 days after discharge were counted as having a single readmission. We included covariates that Medicare uses for risk adjustment under the HRRP: age, sex, and a set of condition-specific comorbidities (identified using International Classification of Diseases, Ninth Revision [ICD-9] procedure and diagnosis codes from the index discharge).

We selected hospital characteristics that describe less-modifiable "structural" features to better understand if luck affects some hospitals more than others. These included ownership (not-for-profit, for-profit, or government), teaching status (major teaching, which are hospitals that are members of the Association of American Medical Colleges Council of Teaching Hospitals, minor teaching, which are hospitals with other teaching affiliations/accreditations, and non-teaching hospitals), number of beds (<100, 100–199, 200–399, and ≥400), location (non-metropolitan, metropolitan), census region (Northeast, South, Midwest, West), safety-net status (based on the hospital's Disproportionate Share Hospital (DSH) payment percentage, which refers to the relative share of means tested care they delivered compared to their total; yes, which includes hospitals with 80% or more, no, which are hospitals with less than 80%), and Medicare's share of annual hospital admissions (<40%, ≥40%).

## Analysis

We conducted all analyses using SAS version 9.4 and Stata version 14 over the period 2018–19. University of Minnesota IRB granted approval for this study. Patient informed consent was not required given the data was de-identified.

The goal of our analysis was to compare a hospitals' penalty in FY2015 to penalties in hypothetical fiscal years. We began by recreating HRRP penalties in FY2015 using data on comorbidities from the index admission. We could not use the actual penalties assigned by Medicare in FY2015 because we needed to make sure that we used the same model to assign FY2015 penalties and generate penalties for the hypothetical fiscal years. We could not perfectly replicate Medicare methods because we only had data on comorbidities from the index admission, whereas Medicare used data on comorbidities from both the index admission and inpatient/outpatient visits prior to the index admission. However, to confirm the accuracy of the recreated penalties, we calculated correlation coefficients between hospitals' recreated penalties and their actual penalties as reported by Medicare and found them to be highly correlated (correlation coefficient was 0.81; p < .001). We also found that the distribution of hospital characteristics by penalty status was also similar for actual versus recreated penalties (Table 1). For ease of expression, we refer to recreated penalties as FY2015 penalties. Below we describe our methods for estimating penalties in FY2015 and simulated penalties in hypothetical fiscal years.

**Recreated HRRP penalties for FY2015.** Medicare uses Excess Readmission Ratios (ERRs) to determine whether to assign HRRP penalties and, if penalized, the size of the penalty. We used Medicare's methods to recreate the ERRs and derive HRRP penalty status and total penalty size. This involved several steps.

First, we identified eligible index admissions and readmissions from July 2010 through June 2013 for each hospital and condition. We estimated the predicted readmission rate for

**Table 1. Hospital characteristics by actual and recreated penalty status for FY2015\*.**

| Hospital Characteristics | Actual Penalty Status | | Recreated Penalty Status | |
|---|---|---|---|---|
| | No Penalty | Penalty | No Penalty | Penalty |
| Hospitals, No. | 527 | 2,483 | 636 | 2,374 |
| Ave. Hospital Beds, No. (sd) | 174 (187) | 232 (213) | 165 (175) | 237 (216) |
| Ownership Status (%) | | | | |
| Non-profit | 60.5% | 62.1% | 59.6% | 62.4% |
| For profit | 24.3% | 22.3% | 22.8% | 22.6% |
| Government | 15.2% | 15.6% | 17.6% | 15.0% |
| Teaching Status (%) | | | | |
| Non-Teaching | 68.5% | 62.0% | 67.9% | 61.9% |
| Minor | 27.7% | 29.4% | 28.9% | 29.1% |
| Major | 3.8% | 8.6% | 3.1% | 9.0% |
| Metro Location (%) | 68.1% | 72.8% | 63.5% | 74.2% |
| Census Region (%) | | | | |
| Northeast | 8.2% | 17.2% | 7.7% | 17.7% |
| Midwest | 27.5% | 22.8% | 26.1% | 23.0% |
| South | 36.1% | 42.9% | 37.1% | 42.9% |
| West | 28.3% | 17.1% | 29.1% | 16.4% |
| Safety-Net Hospital (%) | 11.2% | 22.0% | 12.6% | 22.1% |
| Medicare Share | 39.0% | 39.4% | 39.0% | 39.4% |

Note: There were 3,130 eligible hospitals; 120 (3.8%) were missing data on hospital characteristics.

\* We replicated methods used by Medicare to recreate the HRRP penalty status (see Analysis Section 1).

each hospital and condition by taking the mean of the probability of readmission for each of their discharges with a given condition. Readmission probabilities were estimated with a linear probability regression model of unplanned 30-day readmission on HRRP risk-adjustment variables (age, sex, and comorbidities) and hospital-level random effects. We employed linear probability models to minimize the computational burden and time involved with running non-linear models.

Next, we estimated the expected readmission rate for each hospital and condition using another set of linear probability regression models that regressed unplanned 30-day readmission on the HRRP risk-adjustment variables but not hospital-level random effects. For each hospital and condition, we calculated the ERR by dividing the predicted readmissions by the expected readmissions.

Finally, we classified any hospital with an ERR of greater than 1 for any target condition as penalized and estimated the size of each hospital's total penalty as a percentage of the hospital's total Medicare payments. This involved calculating a payment-weighted average ERR across the target conditions for each hospital. Because the maximum HRRP penalty was 3% in FY2015, we capped penalty sizes at this amount and thus penalties ranged from no penalty or 0% to 3%.

**Simulated penalties for 1,000 hypothetical fiscal years.** To examine how random variation would affect the distribution of HRRP penalties over 1,000 hypothetical fiscal years, we created a set of 1,000 randomly assigned, simulated ERRs for each eligible hospital and target condition. This involved four steps.

First, we used regression models to predict the risk of readmission for each eligible index discharge as a function of condition, length of stay (i.e., number of inpatient nights), age group (65–69, 70–74, 75–79, 80–84, 85–89, 90+), sex, dual-eligibility for Medicaid, race/ethnicity (white, black, Asian, Hispanic, other race), season of discharge month (winter, spring, summer, fall), condition-specific co-morbidities (see S1 –S5 Tables in S1 File), and hospital-level fixed effects. In these analyses, we modeled the risk of readmission not only as a function of the HRRP risk adjusters, but also as a function of other observable patient characteristics such as dual eligibility and race.

Next, for each hospital, we drew a random sample of the hospital's index discharges with replacement, that is, we created a bootstrap sample of discharges. The number of discharges in the random sample was equal to the actual number of discharges from the hospital for each target condition from July 2010 through June 2013. These first two steps, taken together, were intended to capture hypothetical random variation in which patients were admitted to the hospital. For example, a hospital will have a higher risk of receiving penalties if it received patients who had higher risk of readmission based on observable patient factors such as race or income that were not included in HRRP risk adjustment in FY 2015.

Next, for each randomly sampled index discharge, we generated a random number between 0 and 1 using a uniform distribution. If the random number (e.g., 0.18) was less than the predicted risk of readmission for the target condition (e.g., 0.20, from the first step above), we counted this as a "simulated readmission." This step captured hypothetical random variation in events occurring after the index discharge.

Finally, we used the simulated readmissions to calculate the simulated ERR for each hospital and condition, assigned a simulated penalty status, and calculated a simulated total penalty size for FY2015, following the same methods as for the recreation of the FY2015 penalties. We replicated the above steps 1,000 times for each hospital, representing 1,000 hypothetical fiscal years during which penalties could be assigned. To correct for additive bias (i.e., "bootstrap bias") in our simulated penalties introduced from repeatedly sampling from the observed data, we added a constant to each simulated penalty equal to the difference between the mean of the

hospital's simulated penalties and their actual penalty assignment This had the effect of centering each hospital's distribution of simulated penalties on the actual penalty assigned by HRRP.

To isolate random variation from events after the index discharge only, we created a second set of simulated penalties representing 1,000 hypothetical fiscal years. To eliminate random variation from which patients were admitted to a hospital, we modified the second step in developing the simulated penalties by using each hospital's actual index discharges for each condition, rather than creating a bootstrap sample. The methods were otherwise the same.

**Empirical analysis of recreated and simulated penalties.**   First, we compared the characteristics of hospitals that were penalized versus not penalized (based on our recreated penalty status).

Then, we graph the distribution of simulated penalties for a particular hospital across 1,000 hypothetical fiscal years. The graph illustrates the shape of the distribution and shows the range of penalties.

To measure discordance between FY2015 penalties and simulated penalties, we determined the proportion of simulations in which the simulated penalty status differed from the FY2015 penalty status, stratified by the FY2015 penalty status. Among hospitals not penalized in FY2015, we found the average penalty across simulations where the hospital was penalized (i.e., "new" penalties). In addition, we found the average actual penalty in FY2015 across simulations for penalized hospitals where the simulated penalty was zero (i.e., "averted" penalties). We separately performed these calculations for simulations that allowed for random variation in both index admissions and probability of readmission (i.e., the "overall" role of luck) and for simulations that allowed for random variation just in the probability of readmission (the role of luck pertaining to events after discharge).

Note that, by design, within-hospital variation in simulated penalties across 1,000 hypothetical fiscal years is due to luck or random variation alone. Thus, to quantify the overall role of luck in HRRP penalties, we calculated average within-hospital measures of dispersion (standard deviation, range, and interquartile range) in penalty size across the 1,000 hypothetical fiscal years. Here too, we generated estimates incorporating random variation in both admissions and probability of readmission and separate estimates just allowing for random variation in readmissions.

In contrast to within-hospital variation in simulated penalties, between-hospital variation in penalties reflects variation in penalties due to luck as well as other factors such as quality of care. Therefore, as another measure to quantify the role of luck in hospital readmission penalties, we decomposed the variance by within-hospital variation in penalties (which represents luck only) and between-hospital variation in penalties (which represents luck and other factors). We assessed the size of the within-hospital variation relative to the between-hospital variation as a proportion.

To assess the association between hospital characteristics and the role of luck, we performed multivariate regressions in which the unit of analysis was a hospital and the dependent variable was the standard deviation of the simulated penalty size for each hospital. Independent variables were hospital ownership, teaching status, size category, metropolitan location, census region, safety net status, and Medicare share. Due to missing values for one or more characteristics, we omitted 3.8% of hospitals from this analysis. We employed Huber-White robust standard errors.

## Limitations

This study has some limitations. First, while our recreated penalties for FY2015 were very close to the penalties actually assigned, our replication was not perfect. The fact that we only

had data on comorbidities from the index hospitalization, whereas Medicare used data on comorbidities from both the index admission and prior outpatient visits, likely contributed to the differences between our estimates and Medicare estimates. However, this should not play an important role in the findings as both the recreated FY2015 penalties and the simulated penalties for the 1,000 hypothetical fiscal years used the same methods for determining penalties. Second, bootstrap sampling might underestimate variation in patient population, especially for smaller hospitals [7]. Third, when we estimate how random variation in which patients were admitted to the hospital influences a hospitals' readmission penalties, we were only able to quantify the role of observable patient characteristics such as race and dual-eligibility for Medicaid that are not included in HRRP risk adjustment but influence the risk of readmission. We are unable to quantify the role of unobservable patient characteristics.

## Results

### HRRP penalties

Out of 3,130 eligible acute care hospitals, our methods estimated that 2,374 (76%) were penalized in FY2015 (**Table 1**). The correlation coefficient between our recreated HRRP penalty status and the actual penalty status reported by Medicare was 0.81. The distribution of hospital characteristics by recreated versus actual penalty status was also similar. Consistent with prior literature, hospitals receiving HRRP penalties (based on our re-creation or actual penalty status) were larger and more likely to be major teaching hospitals, safety-net hospitals, located in metropolitan areas, and located in the Northeast or South census regions [8].

### Overall role of luck in HRRP penalties

There was a large dispersion of penalty size within hospitals. To illustrate this, **Fig 1** plots the cumulative distribution of penalty size within a non-penalized and penalized hospital (chosen at random) across 1,000 hypothetical fiscal years.

Panel A shows the distribution for the non-penalized hospital, which had had roughly 600 patients discharged with diagnoses for HRRP target conditions over the three year HRRP measurement period. The red line shows the distribution when we randomly vary both the patients admitted to a hospital and the readmission status of patients conditional on hospital discharge. The blue line shows the distribution when we only allow random variation in readmission status of patients across the 1,000 fiscal years. We see that the distributions look similar suggesting that random variation in which patients are admitted to a hospital does not play an important role in determining the variation in penalty size within a hospital. For this hospital the standard deviation in penalty size was 0.25 percent of total Medicare payments, signifying sizeable variation in penalty size relative to the maximum penalty allowed. For example, across hypothetical fiscal years the penalties ranged from 0 percent to 3 percent of total Medicare payments. As noted above, this hospital was not penalized in FY 2015, however this hospital was penalized in about 20.3% of hypothetical fiscal years and the average size of the penalty when penalized was 0.32% of Medicare's total payment (0.3 of 3.0 percent max, or 1/10[th] of the maximum penalty allowed). Panel B shows a penalized hospital, where the blue and red overlapping lines again suggesting which patients are admitted to a hospital does not play an important role in the variation of penalty size. The standard deviation in penalty size for this hospital was 0.45, where they were penalized in all but 15% of our hypothetical fiscal years with an average penalty size of .56 (or roughly 1/5[th] of the maximum penalty allowed).

Next, we repeat the analysis for all HRRP eligible hospitals. Across all hospitals in the analysis (including non-penalized hospitals), the average simulated penalty size was 0.78 percent of Medicare payments (1/4[th] of the maximum penalty allowed), while the average standard

## A. Not Penalized Under the HRRP

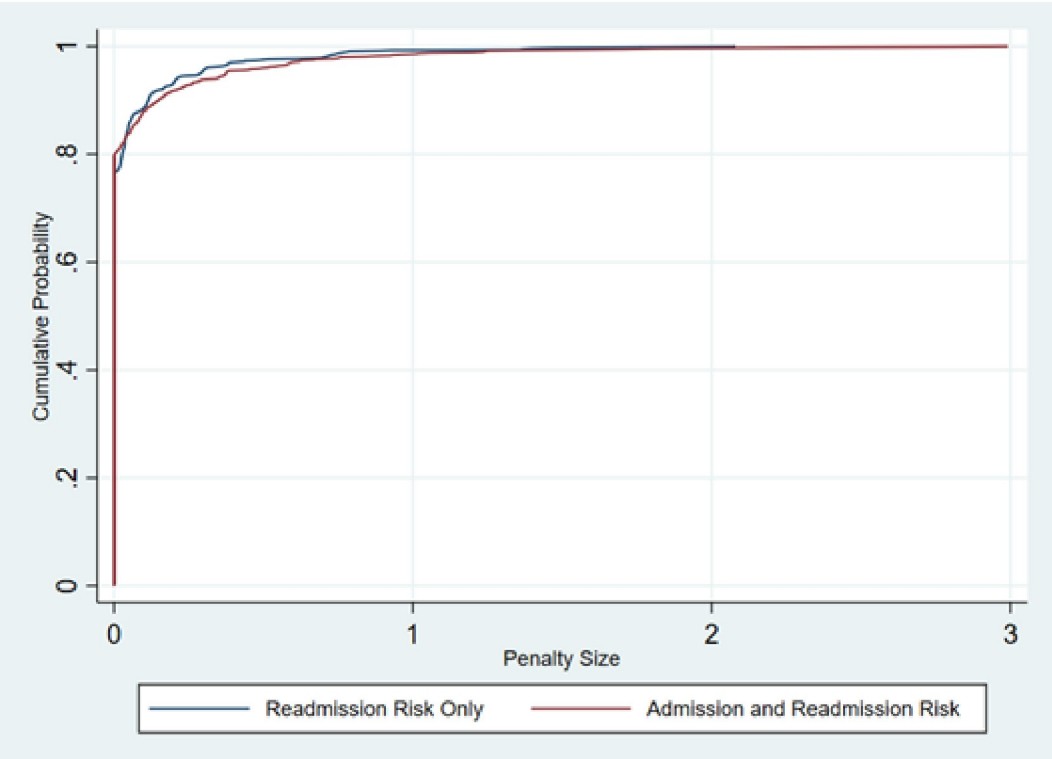

## B. Penalized Under the HRRP

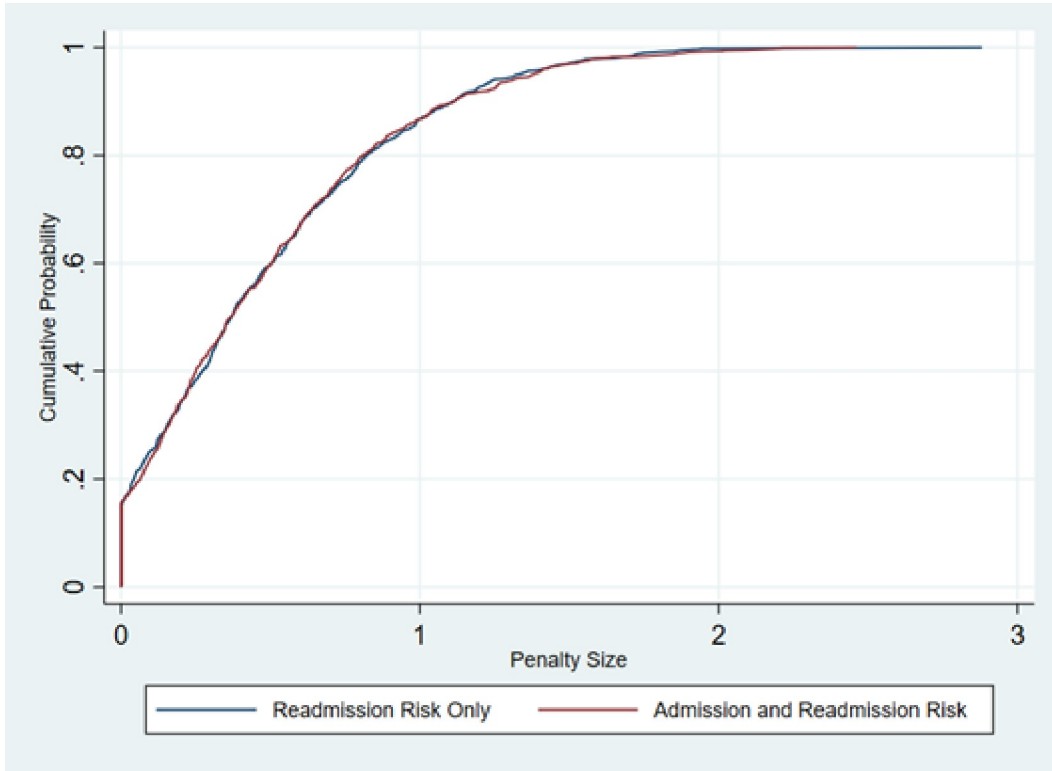

**Fig 1. Cumulative distribution of penalty size for a randomly selected hospital.**

**Table 2. Comparison of recreated HRRP penalty status for FY2015[*] with simulated penalties for 1,000 hypothetical fiscal years.**

| Role of Luck | Penalty Status for FY2015 | Simulated Penalty Status in 1,000 Hypothetical FY | | When Penalty Status Is <u>Discordant</u>, Size of New or Averted Penalty |
|---|---|---|---|---|
| | | Not Penalized | Penalized | Mean PP (SD)‡ |
| *Overall*† | Not Penalized | 76.8% | **23.2%** | + 0.57 Percent of Medicare Spending (0.78) |
| | Penalized | **21.3%** | 78.7% | - 0.43 Percent of Medicare Spending (0.60) |
| *Events after Discharge*† | Not Penalized | 76.9% | **23.1%** | + 0.57 Percent of Medicare Spending (0.78) |
| | Penalized | **21.2%** | 78.8% | - 0.43 Percent of Medicare Spending (0.60) |

[*] We replicated methods used by Medicare to recreate the HRRP penalty status (see Analysis Section 1).

† For the overall role of luck, simulations addressed random variation in the selection of admitted patients to each hospital as well as from events after discharge. The second set of simulations only addressed random variation from events after discharge.

‡ PP, percentage point change in total Medicare payments to a hospital. FY, fiscal year.

deviation of penalty size was 0.89 percent of Medicare payments. A hospital that was not penalized in FY2015 had a 23% chance of being penalized in each of the 1,000 hypothetical fiscal years (**Table 2**, top row).

When this discordance occurred, the average difference in penalty size was 0.57 percent of Medicare payments. Conversely, a hospital that was penalized in FY2015 had a 21% chance of not being penalized in each of the 1,000 hypothetical fiscal years. With such discordance, the average size of the averted penalty was equivalent to 0.43 percent of Medicare payments. The average within-hospital range in penalties across 1,000 simulations was 2.5 percent of Medicare payments and the average interquartile range was 0.7 percent of Medicare payments (**Table 3**).

Recall that in contrast to within-hospital variation in simulated penalties, across hospital variation in penalties reflects variation in penalties due to luck as well as other factors such as quality of care. Therefore, as another measure, to quantify the role of luck in hospital readmission penalties, we decomposed the variation in simulated penalties to compare within-hospital variation in penalties (which represents luck only) to across-hospital variation in penalties (which represents luck and other factors) and found that variance of penalties within hospitals were 82% of variance in penalties across hospitals.

## Role of luck from events after the index discharges

Measures of discordance between FY2015 and simulated penalties were very similar in the second set of simulations. For example, a hospital that was not penalized in FY2015 had a 23% chance of being penalized in each of the 1,000 hypothetical fiscal years and, when this discordance occurred, the new penalty equated to 0.57% of its total Medicare payments (**Table 2**, lower portion). Similarly, a hospital that was penalized in FY2015 had a 21% chance of not

**Table 3. Summary of measures of dispersion of penalty size within a hospital.**

| | Overall† | Events after Discharge† |
|---|---|---|
| Standard Deviation | 0.54 percentage points | 0.54 percentage points |
| Range | 2.45 percentage points | 2.45 percentage points |
| Interquartile Range | 0.68 percentage points | 0.67 percentage points |

† For the overall role of luck, simulations addressed random variation in the selection of admitted patients to each hospital as well as from events after discharge. The second set of simulations only addressed random variation from events after discharge.

being penalized in each of the 1,000 hypothetical fiscal years and, when this discordance occurred, the averted penalty represented 0.43% of the hospital's total Medicare payments. The measures of dispersion of penalties were also very similar in the second set of simulations, which isolated random variation from events after the index discharge (Table 3).

Given the similarity between results for the overall role of luck and for the role of luck from events after discharge, we can infer that randomness from who was admitted to a hospital had a small influence on the assignment of penalties.

## Differences in overall role of luck according to hospital characteristics

Several hospital characteristics were associated with a smaller role of luck in HRRP penalties. Luck, as measured by the standard deviation of penalty size, played a smaller role for hospitals with more beds, teaching hospitals, safety-net hospitals, and hospitals outside of the Northeast region (Table 4).

**Table 4. Misclassification and measure of dispersion of penalty size, shown by hospital characteristics.**

| Characteristics | Standard Deviation of Penalty Size | | |
|---|---|---|---|
| | *Coefficient* | *Standard Error* | *P-Value* |
| Ownership Status | | | |
| Non-profit (ref) | | | |
| For profit | 0.0549 | (0.0145) | < .001 |
| Government | 0.00726 | (0.0151) | 0.632 |
| Teaching Status | | | |
| Non-teaching (ref) | | | |
| Minor | -0.0580 | (0.0126) | < .001 |
| Major | -0.108 | (0.0183) | < .001 |
| Hospital Beds | | | |
| Less than 100 (ref) | | | |
| 100 to 199 | -0.0369 | (0.0158) | 0.019 |
| 200 to 399 | -0.141 | (0.0162) | < .001 |
| 400 and more | -0.239 | (0.0195) | < .001 |
| Rural Status | | | |
| Non-Metro (ref) | | | |
| Metro | 0.0169 | (0.0146) | 0.248 |
| Region | | | |
| Northeast (ref) | | | |
| South | -0.0505 | (0.0141) | < .001 |
| Midwest | -0.0326 | (0.0153) | 0.033 |
| West | -0.100 | (0.0175) | < .001 |
| Safety Net Status | | | |
| No (ref) | | | |
| Yes | -0.0480 | (0.0129) | < .001 |
| Medicare Share of Admissions | | | |
| Under 40% (ref) | | | |
| 40% and more | -0.0139 | (0.0119) | 0.243 |
| Constant | 0.681 | (0.0200) | < .001 |
| Observations | 3,010 | | |
| R-squared | 0.144 | | |

Note: Hospitals with any missing variables were removed (3.8%). Huber-White Robust standard errors are shown.

## Discussion

In our analysis based on simulations using national data from Medicare, luck played a significant role in determining the size of hospital penalties. Nearly all of the role that luck played in hospital readmissions and the assignment of penalties arose from events occurring after hospital discharge, rather than from randomness in who is admitted to the hospital. Hospitals with more beds, teaching hospitals, and safety-net hospitals exhibited less random variation in the assignment of HRRP penalties, meaning luck played a smaller role.

Luck playing a significant role in determining HRRP penalties has two important implications. First, because penalties are to a significant extent determined by luck, it is possible that hospitals making equal investments in quality will suffer different penalty sizes, leading some to perceive the policy as unfair. For example, a hospital that has made significant investments in quality improvement might still be penalized, generating a certain degree of disenchantment toward quality improvement. Perceptions of fairness are important for achieving organizational commitment, [9], and basing readmission penalties on chance may reduce enthusiasm for improving it. Indeed, hospital administrators have reported little ability to influence their readmission rates and avoid penalties, and have indicated that lowering readmissions is a lower priority for them [10].

Second, this is an inefficient policy. Penalties signal hospitals on whether to change their practices to lower readmission. If they are based on chance, then low performers may not know when to change and high performers may change when they do not need to. This means that hospital practice changes stemming from the HRRP may not always be an improvement, as some evidence indicates mortality has been higher for heart failure patients since the start of the program [11]. Policy makers need to consider the tradeoff the HRRP appears to be making: are the savings generated from randomly distributing penalties worth the risk of incentivizing changes in care delivery in ways that could adversely impact patient access and welfare.

Prior research has documented that variation in readmission rates is driven by differences in patient case-mix [3]. Important determinants of readmission such as socioeconomic status [12–14] and community disadvantage [15, 16] have not, until recently, been accounted for in HRRP penalty assignment—a shortcoming that has left hospitals serving these communities accountable for their locations' above average readmission risk [17–19]. In response to these criticisms, Medicare began (starting in FY2019) comparing hospitals with similar proportions of disadvantaged patients when calculating HRRP penalties. While these changes could change hospitals' expectations of being penalized, our analysis suggests that these changes will not diminish the role of luck in determining readmission penalties as we find that variation in the patient population does not play an important role in explaining variation in excess readmission risk. Rather, our analysis supports other research that has found post-discharge care patterns play an important role in determining unplanned readmissions, and by not including these factors in their risk-adjustment Medicare leaves open a bigger role for luck in assigning penalties [20, 21].

Our findings are likely relevant for other Medicare value-based payment initiatives such as the Hospital-Acquired Condition Reduction Program and Merit-based Incentive Payment System. In these programs, penalties and bonuses will depend substantially on luck: which patients a particular provider drew versus which patients' other providers drew over the same period in addition to the inherent luck in readmissions risk.

One implication of our findings is that we need to reevaluate the design of HRRP in a way that increases the observation count readmission rates are based on. Some plausible improvements might include: limiting the program to large hospitals, increasing the number of conditions, mixing readmission with other quality outcomes, or increasing the length of the

evaluation period. Discounting penalties by the level of uncertainty around the hospital's observed readmission rate might be another way to lessen the financial consequences from misclassification. These changes may also address that a substantial proportion of the observed reductions in readmissions under HRRP can be attributed to regression to the mean [22].

## Conclusion

Rewards and penalties are frequently used in health care to elicit behavior from providers that improves efficiency of care. Our findings suggest that in programs like the HRRP, all providers run a risk of being penalized based on luck or random chance rather than true clinical quality and some providers face greater risk than others. As pay-for-performance programs mature and become more commonplace, care should be taken to account for the fact that assessed performance may not be based on true clinical quality but based on a chance.

## Supporting information

**S1 File.**
(DOCX)

## Acknowledgments

We would like thank the attendees of the 2018 American Society of Health Economists Meeting and Academy Health's Annual Research Meeting for their helpful comments on the analyses we presented here.

## Author Contributions

**Conceptualization:** Andrew D. Wilcock, José Escarce, Peter J. Huckfeldt, Neeraj Sood.

**Formal analysis:** Andrew D. Wilcock, Sushant Joshi.

**Funding acquisition:** José Escarce, Peter J. Huckfeldt, Teryl Nuckols, Neeraj Sood.

**Investigation:** Andrew D. Wilcock, Sushant Joshi, José Escarce, Peter J. Huckfeldt, Teryl Nuckols, Ioana Popescu, Neeraj Sood.

**Methodology:** Andrew D. Wilcock, Sushant Joshi, José Escarce, Peter J. Huckfeldt, Neeraj Sood.

**Project administration:** Teryl Nuckols, Neeraj Sood.

**Resources:** Neeraj Sood.

**Supervision:** José Escarce, Peter J. Huckfeldt, Teryl Nuckols.

**Writing – original draft:** Andrew D. Wilcock, José Escarce, Peter J. Huckfeldt, Teryl Nuckols, Ioana Popescu, Neeraj Sood.

**Writing – review & editing:** Andrew D. Wilcock, Sushant Joshi, José Escarce, Peter J. Huckfeldt, Teryl Nuckols, Ioana Popescu, Neeraj Sood.

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
