## [Decision Letter · Decision Letter 0]

15 Jul 2021

PONE-D-21-15295

Luck of the Draw: Role of Chance in the Assignment of Medicare Readmissions Penalties

PLOS ONE

Dear Dr. Wilcock,

Thank you for submitting your manuscript to PLOS ONE. After careful consideration, we feel that it has merit but does not fully meet PLOS ONE’s publication criteria as it currently stands. Therefore, we invite you to submit a revised version of the manuscript that addresses the points raised during the review process.

Please address the following journal requirements and additional editor and reviewer comments.

We look forward to receiving your revised manuscript.

Kind regards,

Fernando A. Wilson, PhD

Academic Editor

PLOS ONE

Journal Requirements:

2. Please include your actual numerical p-values in Table 4

3. In your ethics statement in the Methods section and in the online submission form, please provide additional information about the data used in your retrospective study. Specifically, please ensure that you have discussed whether all data were fully anonymized before you accessed them and/or whether the IRB or ethics committee waived the requirement for informed consent. If patients provided informed written consent to have data from their medical records used in research, please include this information

4. Thank you for providing the date(s) when patient medical information was initially recorded. Please also include the date(s) on which your research team accessed the databases/records to obtain the retrospective data used in your study

Additional Editor Comments (if provided):

- Please follow PLOS ONE formatting requirements on exhibits. For example, all tables must be embedded within the text after the paragraph where they are first mentioned. Figure captions must also be embedded within the text.

- In Introduction, in 1-2 sentences, please describe the Medicare program. Please bear in mind that PLOS ONE has an international readership and they may not be highly familiar with US healthcare programs. Similarly, add more detail on HRRP, eg, possible penalties, 30-day “all-cause” readmissions, evaluation period, etc.

- Pg. 5: What is the size of the study population?  State how many hospitals were included and the number and % missing.

- Pg. 6: Write out “CMS” at 1st mention.  Either write out the condition-specific comorbidities or list them in Supporting Information. Clarify how minor vs. major teaching hospitals were defined. “number of beds” instead of “bed size”. Briefly describe DSH. Add “and” in front of “Medicare share” and briefly define it. Finally, clarify why these particular hospital characteristics were selected.

- Pg. 7: Write out “ERR” at 1st mention and define what it is.  

- Pg 8:  Add more detail on the regression variables, eg, length of stay is number of inpatient nights, what were the age groups and seasons? Clarify why the variables were selected for the regression modeling. Patient race is mentioned but not listed in the 1st paragraph nor defined. Was ethnicity included? To clarify, were any other patient characteristics, eg, income or other socio-economic status variables, included?  Finally, as reviewers suggest, please provide more detail on the modeling and report the regression estimates in Supporting Information. 

- Pg. 8: “between 0 and 1”.

- Pg. 9: “Then, we graphed…”. “we calculated the average penalty…”. “In addition, we calculated…”.  Be consistent in using commas for “i.e.,”.

- Pg. 10: add commas around “by design”. Also, here and elsewhere in the text, be consistent in using either “within-hospital” or “within hospital”. Add a comma between “penalties” and “we”.

- Pg. 11: add a comma between “penalties” and “we”. Please specify these SES measures on pg. 8, 1st paragraph.

- Pg. 12: add “and” between “FY2015” and “it”. Extra space in “28.8 %”.

- Pg. 13: use “1,000” to be consistent.

- Pg. 14: “In our analysis…”. “because penalties” instead of “since”.  Here and elsewhere, avoid contractions like “it’s”, “don’t”, etc. Use a comma instead of semi-colon in front of “generating”.

- Pg. 15: avoid starting sentences with “And”. “variation in the patient population…”.

- Pg. 16: add “the” in front of “Hospital-Acquired…”.

- Figure 1: suggest adding the % symbol to the x-axis labels. Also, use consistent terminology in figure title vs. text, ie, “cumulative distribution” vs. “cumulative density function”. The y-axis title in the caption should be embedded in the figure (and the caption must be embedded within the text). Please confirm that PLOS ONE requirements regarding figures have been followed (https://journals.plos.org/plosone/s/figures).

- Conclusions: add a comma between “commonplace” and “care”.

- Please include an Acknowledgements section prior to References. Refer to the following link for more information on manuscript requirements:

https://journals.plos.org/plosone/s/submission-guidelines. 

- References must follow PLOS ONE requirements, eg, citations in text should use square brackets, journal articles must use NCBI abbreviations, #8 has incorrect formatting of author names, etc. Please refer to the following link:

https://journals.plos.org/plosone/s/submission-guidelines#loc-references

- If you include Supporting Information, please follow PLOS ONE requirements in the link below:

https://journals.plos.org/plosone/s/supporting-information

Reviewers' comments:

Reviewer's Responses to Questions

**Comments to the Author**

1. Is the manuscript technically sound, and do the data support the conclusions?

Reviewer #1: Partly

Reviewer #2: Partly

2. Has the statistical analysis been performed appropriately and rigorously? 

Reviewer #1: Yes

Reviewer #2: No

3. Have the authors made all data underlying the findings in their manuscript fully available?

Reviewer #1: Yes

Reviewer #2: Yes

4. Is the manuscript presented in an intelligible fashion and written in standard English?

Reviewer #1: Yes

Reviewer #2: Yes

5. Review Comments to the Author

Reviewer #1: This study used 2010-2013 Medicare claims data and a simulation model to examine the role of luck on 30-day unplanned readmission among the Medicare beneficiaries whose primary diagnoses during index admission (IA) were under the Hospital Readmission Reduction Program. The authors found that luck played a significant role in hospitals’ readmission performance.

[1] Major

1. The authors may need clearer logic to justify their assumptions regarding the two mechanisms through which luck would affect hospital readmission.

Although the authors emphasized the role of luck (i.e., random variations), their study design did not clearly differentiate (i) real randomness, as demonstrated in their mechanisms, from (ii) the determinants of IA (e.g., various health services utilization prior to 30 days of IA) and readmission (e.g., number of procedures during the IA, discharge status of IA) that were identified in relevant literature and available in the Medicare claims data but were not adjusted for their study.

For example, more than two thirds of readmissions occurred for the different primary diagnoses from the IA, but a substantial proportion of patients used various post-discharge care related with these diagnoses, which significantly decreased readmission (e.g., Dharmarajan et al., JAMA, 2013; Tak et al., JGIM, 2019).

2. In Analysis (pp.7-8), the authors simply stated that they used “regression models” to estimate the predicted number of readmissions for each hospital and risk of readmission for each eligible index discharge. Please specify what regression models the authors used.

[2] Minor

1. Briefly explain the criteria for the eligible IA (e.g., a discharge status of not being transferred to another acute care facility) and for the 30-day all-cause unplanned readmission (e.g., first readmission only when there were multiple readmissions within 30 days), and cite references.

2. Please write the p-values for the correlation coefficients (and other statistical analyses, if applicable).

3. Please spell out what ERR stands for.

Reviewer #2: The authors present interesting results from a simulation study of Medicare readmission penalties. Their simulations were designed to assess two sources of randomness in readmission: 1) who gets admitted to the hospital in the first place; and 2) post-discharge occurrences (for who gets readmitted). Authors call this randomness "luck" and assess how luck impacts the penalties that get imposed. They find that most of the variability in penalties can be attributed to the randomness in events after discharge. They also found that larger hospitals, teaching hospitals, safety-net hospitals and hospitals with a larger share of Medicare patients have less variability in penalties (so a lower contribution of luck). The manuscript will be strengthened if the authors consider the following points.

1. Authors use ANOVA to assess the between versus within hospital variability. Based on the author's description, it is not entirely clear what is being done here. Is the penalty the outcome with factors of hospital and maybe hypothetical fiscal year? Are the assumptions of ANOVA met?

2. Authors use multivariate regression to assess how different factors might be associated with "luck", quantified by the standard deviation in the penalties across the 1000 hypothetical fiscal years for each hospital. Were the assumptions of the linear regression model met for these data? Were there any issues of collinearity (or substantial overlap) among the factors considered in the model?

3. In Figure 1, authors show the cumulative distribution function of the penalties across the 1000 hypothetical fiscal years for a randomly chosen hospital, which happens to be a hospital that was not actually penalized. For comparison, authors should also include a figure for a randomly chosen hospital that was penalized to see the distribution of hypothetical penalties for a penalized hospital.

4. Authors state early on in the results that hospitals that are more likely to be penalized are major teaching hospitals, safety-net hospitals, those located in metropolitan areas, and those in the Northeast or South census regions, though many of these factors are associated with lower standard deviation in penalties (at least major teaching hospitals, safety-net hospitals, and the South census region), indicating less of an impact of luck. Do authors have any comment about this?

Minor points:

1. On the top of page 7, authors use "ERR" without defining it first.

2. For the 1st step in the simulated penalties for 1000 hypothetical fiscal years, authors say they use regression models. I'm assuming the outcome is readmission (yes/no) in these models, so to be more clear in what was done, authors should say "logistic regression models".

3. On the top of page 11 "The fact we" should possibly be "The fact that we".

4. In Table 1, authors provide the average number of beds for the non-penalized and penalized hospitals (actual and recomputed). Even though the sample size is quite large, it still would be useful to also include the standard deviation in the number of beds.

5. On page 12, end of the 3rd line of the 1st paragraph, change "it" to "and" or rephrase the sentence.

6. Also on page 12, about half way down the page, authors talk about the standard deviation in penalty size for the hospital illustrated in Figure 1 as "significant variation". How do authors define "significant variation"?

7. In the next paragraph (last paragraph on page 12), authors mention the average penalty size across all hospitals - is this the average penalty size when penalized?

8. On the bottom of page 15, (2nd line from the bottom), change "variation is" to "variation in"

9. In Table 4, for the 3rd column header, change "Standard deviation" to "Standard Error", since I'm assuming that is what is being presented (the standard error of the coefficient).

6. PLOS authors have the option to publish the peer review history of their article (what does this mean?). If published, this will include your full peer review and any attached files.

Reviewer #1: No

Reviewer #2: No

---

## [Author Response · Author response to Decision Letter 0]

16 Oct 2021

We have uploaded a word document "Responses" that includes our response to each of the editor and reviewer comments we received.

---

## [Editor Report · Decision Letter 1]

1 Dec 2021

Luck of the Draw: Role of Chance in the Assignment of Medicare Readmissions Penalties

PONE-D-21-15295R1

Dear Dr. Wilcock,

We’re pleased to inform you that your manuscript has been judged scientifically suitable for publication and will be formally accepted for publication once it meets all outstanding technical requirements.

Kind regards,

Fernando A. Wilson, PhD

Academic Editor

PLOS ONE
---

## [Editor Report · Acceptance letter]

6 Dec 2021

PONE-D-21-15295R1 

Luck of the Draw: Role of Chance in the Assignment of Medicare Readmissions Penalties 

Dear Dr. Wilcock:

I'm pleased to inform you that your manuscript has been deemed suitable for publication in PLOS ONE. Congratulations! Your manuscript is now with our production department. 

Kind regards, 

on behalf of

Dr. Fernando A. Wilson 

Academic Editor

PLOS ONE